# Virtual Screening of Soybean Protein Isolate-Binding Phytochemicals and Interaction Characterization

**DOI:** 10.3390/foods12020272

**Published:** 2023-01-06

**Authors:** Panhang Liu, Annan Wu, Yi Song, Jing Zhao

**Affiliations:** 1College of Food Science and Nutritional Engineering, China Agricultural University, Beijing 100083, China; 2China National Engineering Research Center for Fruit & Vegetable Processing, Beijing 100083, China; 3Beijing Key Laboratory for Food Non-Thermal Processing, Beijing 100083, China

**Keywords:** soybean protein isolate, phytochemicals, interaction, molecular dynamics simulation

## Abstract

Soybean protein isolate (SPI) and small molecule interactions have drawn more and more attention regarding their benefits for both parts, while research on large-scale investigations and comparisons of different compounds is absent. In this study, a high throughput virtual screening was applied on a phytochemical database with 1130 compounds to pinpoint the potential SPI binder. Pentagalloylglucose, narcissoside, poliumoside, isoginkgetin, and avicurin were selected as the top-five ranking molecules for further validation. Fluorescence quenching assays illustrated that isoginkgetin has a significantly higher apparent binding constant (*Ka*) of (0.060 ± 0.020) × 10^6^ L·mol^−1^, followed by avicularin ((0.058 ± 0.010) × 10^6^ L·mol^−1^), pentagalloylglucose ((0.049 ± 0.010) × 10^6^ L·mol^−1^), narcissoside ((0.0013 ± 0.0004) × 10^6^ L·mol^−1^), and poliumoside ((0.0012 ± 0.0006) × 10^6^ L·mol^−1^). Interface characterization by MD simulation showed that protein residues E172, H173, G202, and V204 are highly involved in hydrogen bonding with the two carbonyl oxygens of isoginketin, which could be the crucial events in SPI binding. Van der Waals force was identified as the major driven force for isoginketin binding. Our study explored SPI–phytochemical interaction through multiple strategies, revealing the molecular binding details of isoginkgetin as a novel SPI binder, which has important implications for the utilization of the SPI–phytochemical complex in food applications.

## 1. Introduction

Soybean protein isolate (SPI) is a widely used plant-based protein containing all the essential amino acids needed to fulfill adults’ nutritional requirements. In addition, SPI exhibits many biological activities including decreasing serum lipids, improving intestinal microbiota, modulating immunity, etc. [1,2,3]. Although widely used in meat products and others [4], SPI has some unfavorable characteristics affecting its application, such as low solubility, poor emulsifiability, etc. [5]. Physical, chemical, and enzymatic modification methods had been utilized to improve the functional properties of SPI. However, physical and chemical methods have low specificity and enzymatic methods tend to produce unfavorable bitter peptides [6]. Recently, molecular interaction between proteins and small molecules has been developed as a new strategy to improve the functional properties of proteins, especially for plant-based proteins [7]. Two types of interactions are demonstrated between proteins and small molecules, namely, covalent and non-covalent interactions. Non-covalent bonding can be achieved through hydrogen bonds, van der Waals forces, and hydrophobic interactions [8]. For example, (-)-epigallocatechin-3-gallate (EGCG) noncovalently binds SPI and increases the foaming, emulsifying, and antioxidant properties of SPI [6]. Rutin significantly increases the emulsifying properties of myofibrillar protein [9] and improves the foaming properties of soybean protein [10]. Alternatively, covalent bonding can be achieved by enzymatic, thermal, or alkaline methods. Covalent binding formed between catechins and whey protein changed the protein secondary and tertiary structure and the conjugate improved foaming and emulsifying properties [11]. However, due to undesirable side products formation such as quinone polymers and intense operating conditions such as high temperature and high pH in covalent interaction [12], non-covalent modification is more preferred and environmentally friendly for utilization in the food industry. For example, the composite of β-lactoglobulin and EGCG was used to manufacture stable Pickering emulsions carrying bioactive compounds [13]. EGCG was also used to alter the surface hydrophobicity of thermally induced SPI to improve the gel strength, water-holding capacity, and rheological properties of SPI-based gels [14]. Overall, small molecule non-covalent binding has become a promising strategy for protein modification.

To explore more compounds with potential SPI-binding capacity, SPI–small molecule interactions need to be investigated on a larger scale and with multiple strategies. Phytochemicals have drawn more and more attention in modern food science due to their unique bioactivity and structural diversity [15,16,17,18,19]. Many phytochemicals have been reported for their protein-binding capacities, in which polyphenolic compounds are the most studied, such as catechins, resveratrol, curcumin, et al. Until now, research on the characterization of SPI–phytochemical interactions is still limited [16]. In this study, a high throughput virtual screening was applied to a phytochemical database consisting of 1130 compounds to pinpoint the potential SPI-binding phytochemicals. Compounds with the lowest docking free energies were validated by in vitro fluorescence quenching assays. MD simulation was applied to the best binder to interpret the interaction details. Our work compared the binding preference of SPI to different phytochemicals, which has important implications for the utilization of natural small molecules in the fabrication of SPI conjugates utilized in the food industry.

## 2. Materials and Methods

### 2.1. Materials

Soy protein isolate (90.0% protein) was purchased from Shandong Biological Products Co., Ltd. (Linyi, China). Pentagalloylglucose, Narcissoside, Poliumoside, Isoginkgetin, and Avicurin (purity > 90%) were purchased from Chengdu Push Bio-technology Co., Ltd. (Wuhou, China). All other chemicals and reagents were of analytical grade. Syringe filters were purchased from Sigma-Aldrich (St. Louis, MO, USA).

### 2.2. Virtual Screening of SPI-Binding Phytochemicals

The binding pocket on 11S glycinin (PDB ID: 1OD5) was determined in previous research [5]. Briefly, Schrödinger SiteMap (Schrödinger Release 2021-3, NewYork, NY, USA) was used to predict the binding pockets on soybean 11S glycinin. As a well-defined SPI binder, epigallocatechin gallate (EGCG) was used as a model compound and docked into the predicted pockets. The grid files were generated by the Receptor Grid Generation module and the docking was evaluated by the ligand docking function. The EGCG docking scores were used to select the optimal binding pocket. Then, a virtual screening was performed on a phytochemical database with 1130 compounds. We performed docking using Schrödinger Glide, where small molecules were placed separately into previously identified docking pockets [5], then calculated the docking score and root mean square deviation (RMSD). RMSD of each atom is used to resolve the molecular conformation change deviation. If RMSD values are less than 2, reproducing the protein–ligand interaction is considered to be successful.

### 2.3. Validation of SPI-Phytochemicals Binding by Fluorescence Quenching Assays

Binding of SPI and high-scored phytochemicals was characterized using a fluorescence quenching assay. Specifically, a series of concentration gradients of small molecules were mixed with protein solutions and added to the 96-well plate, with the final protein concentration maintained at 0.135 mg/mL. Excitation light at 280 nm was used and the emission spectra were scanned in the wavelength range of 310–450 nm.

The fluorescence in-filter effect is corrected using the following equation:
(1)$$\begin{array}{c} F_{cor}(\lambda_{ex},\lambda_{em})=F_{obs}(\lambda_{ex},\lambda_{em})\;CF_{p}(\lambda_{ex})\;CF_{s}(\lambda_{em})\\ \approx F_{obs}(\lambda_{ex},\lambda_{em})10^{(A_{em}+A_{ex})/2} \end{array}$$
where *CF_P_* and *CF_S_* are the correction factors for excitation and emission light, respectively, *A_ex_* and *A_em_* are the absorbances of emission and excitation light, *F_obs_* is the observed fluorescence intensity, and *F_cor_* is the corrected fluorescence intensity [20].

The Stern–Volmer equation was used to calculate the quenching constant:*F*_0_*/F* = 1 + *K_sv_*[*Q*].(2)

*F* and *F*_0_ are fluorescence intensity with and without phytochemicals in the protein solution, respectively. *K_sv_* is the quenching constant, and [*Q*] is the concentration of phytochemicals. Equation (2) was used to estimate the binding parameters:*lg[(F*_0_*− F)/F]* = *lgK_a_* + *nlg*[*Q*].(3)

*K_a_* is the apparent binding constant, and *n* is the number of binding sites.

### 2.4. Molecular Dynamics Simulation

All MD simulations were performed using GROMACS 2021 [21] with Amberff99SBildn force field. The initial structures of 11S glycinin used in the MD simulation were derived from the PDB structure (ID:1OD5). The molecular structure parameters of isoginkgetin were obtained from ChemSpider and optimized by ORCA 5.0 [22] and Multiwfn [23], then Sobtop [24] was used to establish the GAFF force field. The complex was prepared by Pymol, using the conformation obtained by docking as the initial conformation [25]. First, SPI and isoginkgetin were placed in a dodecahedron box with a volume of 455.86 nm^3^ and the box was filled with TIP3P water molecules. To balance the charge, 13 Na^+^ were added to the system. Then, the system was energy optimized to convergence. Temperature was stabilized near 300 °C using NVT at 200 ps and pressure was stabilized near 1 bar using NPT at 1 ns. Finally, the limiting potential of the system was lifted and the simulation was performed for 50 ns.

Throughout simulations, NVT uses V-rescale coupling and NPT uses Parrinello–Rahman coupling with a time step of 2 fs. Van der Waals forces and short-range electrostatic interactions are truncated at a radius of 14 Å. The long-range electrostatic interactions are calculated using the particle mesh Ewald (PME) method. The root mean square deviation (RMSD) of the SPI skeleton, the radius of gyration (*R_g_*) of the SPI, and the average distance between the SPI and ISO were calculated using the program that comes with gromacs 2021. Gmx_MMPBSA is a program based on Amber’s MMPBSA.py [26]. We can use it to calculate the complex’s binding free energy and the binding contribution of each amino acid residue using the molecular mechanics Poisson–Boltzmann surface area (MMPBSA) method.

### 2.5. Statistical Analysis

The statistical analysis software GraphPad (San Diego, CA, USA) was used to analyze the data. Comparisons of means were determined by Duncan’s test at the 5% significance level using a one-way analysis of variance (ANOVA).

## 3. Results and Discussion

### 3.1. Virtual Screening of Phytochemicals against Soybean Protein

A phytochemical library containing 1130 compounds was used for virtual screening against the binding pocket on 11S glycinin (PDB ID: 1OD5) determined in previous research [5]. The top 20 compounds are listed in Table 1. The results show that pentagalloylglucose exhibits the lowest binding energy (−9.91 kcal/mol), followed by narcissoside, poliumoside, isoginkgetin, avicularin, which are −8.222, −8.183, −7.984, −7.984 kcal/mol, respectively (Figure 1). It is worth noticing that the top 5 compounds are all glycoside-substituted polyphenols except for isoginkgetin. The stereo views of the selected 11S-phytochemical complex are shown in Figure 1. Detailed interactions are shown in Figure 2. The major amino acids involved in pentagalloylglucose binding include Arg161, His173, Met177, Glu200, and Gly202 (Figure 2A). For narcissoside, amino acids including Glu172, Thr176, Glu200, and Gly202 dominate the binding (Figure 2B). For poliumoside (Figure 2C), Glu172, Thr176, Glu200, and Gly202 are highly involved. For isoginkgetic, Glu172, His173, and Ser206 are involved in the interaction (Figure 2D). As for avicularin, it mainly interacts with Arg161, Glu172, His173, and Gly202 (Figure 2E). The results indicate that electrically charged amino acids (His and Arg with positive charge and Glu with negative charge) and polar amino acids (Thr and Ser) are highly involved in the phytochemical binding. It indicates that electrostatic interaction and hydrogen bonding could be the driving forces for SPI–phytochemical binding.

### 3.2. Characterization of SPI-Phytochemical Interaction by Fluorescence Quenching Assays

Fluorescence quenching assay is widely used in characterizing protein–ligand interactions. Fluorescence quenching is a process that lowers the fluorescence intensity of protein because of interaction with quencher molecules. A lot of small molecules have been proven to be fluorescence quenchers. In this study, the binding of top phytochemicals including pentagalloylglucose, narcissoside, poliumoside, isoginkgetin, and avicularin to SPI was investigated based on a fluorescence quenching assay. Fluorescence quenching assay was also carried out on SPI and n-butanol interaction, as a negative control (Appendix A).

1,2,3,4,6-O-pentagalloylglucose (PGG) exhibited the lowest binding free energy in virtual screening (−9.91 kcal/mol, see Table 1). It is found in a wide variety of herbals, including Punica granatum, Mangifera indica, and Elaeocarpus sylvestris. It has been used for cancer prevention because of its antioxidant and antiinflammation properties [27]. Figure 3B shows the fluorescence spectra of SPI at different concentrations of PGG. Fluorescence intensity significantly decreases with increasing PGG concentration (from 10 to 60 μM), indicating the interaction between PGG and SPI. It is worth noting that a red shift (from 337 nm to 340 nm) was observed with the addition of PGG (Figure 3B), implying that the fluorescent group of SPI was in a more hydrophobic environment in the presence of PGG. This could be due to a slight conformational change of SPI induced by PGG [5].

As shown in Figure 3C, *F*_0_/*F* and PGG concentration were fitted well by the Stern-Volmer equation (Equation (1)). The linear Stern–Volmer diagram illustrates that only one type of quenching mechanism occurs (dynamic or static). The quenching constant *K_sv_* was calculated to be 0.047 ± 0.001 × 10^6^ L·mol^−1^ (Table 2). The apparent binding constant *K_a_*, and the average number of binding sites *n* were obtained by fitting experimental data to Equation (2). PGG–SPI interaction has a *K_a_* of (0.049 ± 0.010) ×10^6^ L·mol^−1^ and an average number of binding sites (*n*) as 1.00 ± 0.04.

Narcissoside is the phytochemical with the second lowest binding energy in virtual screening (−8.222 kcal/mol, see Table 1). It is a monomethoxy flavone derivative (Figure 4A), which has been reported to alleviate mitochondrial oxidative stress and have anti-acute myeloid leukemia effects [28]. Like PGG, adding narcissoside also leads to a decrease in the fluorescence intensity of SPI. With increasing concentration, red shift of fluorescence peak happens (from 340 to 345 nm), which indicates that the bonding model of narcissoside is similar to PGG. As shown in Figure 4 and Table 2, the quenching constant *K_sv_* for narcissoside was fitted to be (0.020 ± 0.001) ×10^6^ L·mol^−1^. The apparent binding constant for narcissoside *K_a_* was obtained to be (0.0013 ± 0.0004) × 10^6^ L·mol^−1^, which is 38 times lower than PGG, indicating a significantly lower binding capacity. The average number of binding sites (*n*) was fitted to be 1.71 ± 0.16.

Figure 5 shows the SPI fluorescence quenching by poliumoside, which has a docking energy of −8.183 kcal/mol in the virtual screening. Poliumoside is a phenylethanoid glycosides isolated from *Brandisia hancei* which is an advanced glycation product formation inhibitor and has anti-inflammatory and antioxidant activities [29,30,31,32]. Fluorescence intensity of SPI decreased with the increase in poliumoside concentration (from 10 to 60 µM; see Figure 5Β). Like PGG, A slight red shift (from 340 to 345 nm) was observed with increasing poliumoside concentration, indicating a conformational change. The binding parameters are shown in Table 2. Poliumoside showed a *K_sv_* of (0.0094 ± 0.0010) × 10^6^ L·mol^−1^. The apparent binding constant *K_a_* was calculated to be (0.0012 ± 0.0006) × 10^6^ L·mol^−1^, which is close to poliumoside and much lower than 1,2,3,4,6-O-pentagalloylglucose. The average number of binding sites *n* is 1.52 ± 0.18.

The highest apparent binding constant was observed for isoginkgetin-SPI interaction (Figure 6). Isoginkgetin is a bis-flavonoid compound isolated from Ginkgo biloba, attenuates lipopolysaccharide induced monoamine neurotransmitter deficiency and depression-like behaviors by downregulating the p38/NF-κB signaling pathway [33]. It shows a binding energy of −7.984 kcal/mol in molecular docking. A quenching constant *K_sv_* of (0.042 ± 0.002) × 10^6^ L·mol^−1^ was obtained and the apparent binding constant *K_a_* was calculated to be (0.060 ± 0.020) ×10^6^ L·mol^−1^, much higher than other compounds (Table 2). Besides, it did not perturb the red or blue shift of SPI fluorescence, representing a different binding mode. The average number of binding sites *n* is 0.98 ± 0.12.

Avicularin, a glycoside of quercetin from the leaves of guava, has been reported to possess a variety of biological properties such as anti-inflammatory, anti-allergic, antioxidant, hepatoprotective, anti-tumor activities, and anti-obesity [34,35]. Figure 7B shows the fluorescence emission spectra of SPI with the addition of avicularin. Unlike other compounds, a blue shift was observed, suggesting that the structure of SPI is more compacted in the presence of avicularin, indicating a different mode of conformational change caused by avicularin. As shown in Figure 7C and Table 2, the quenching constant *K_sv_* for avicularin was fitted to be (0.040 ± 0.001) × 10^6^ L·mol^−1^. The apparent binding constant *K_a_* and the average number of binding sites *n* were calculated to be (0.058 ± 0.010) × 10^6^ L·mol^−1^ and 0.90 ± 0.05, respectively.

Overall, fluorescence quenching assays have indicated distinct SPI-binding parameters and protein structural change among the top 5 phytochemicals. Isoginkgetin exhibited the biggest binding constant ((0.060 ± 0.020) × 10^6^ L·mol^−1^), indicating that isoginkgetin has the highest binding affinity to SPI, which is followed by avicularin, pentagalloylglucose, narcissoside, and poliumoside. Thus, to investigate why SPI has such a preference for isoginkgetin, we further carried out an MD simulation to characterize the molecular details in SPI–isoginkgetin interaction.

### 3.3. Molecular Dynamics Simulation on SPI-isoginkgetin interaction

A 50 ns MD simulation was performed on isoginkgetin and soybean 11S glycinin. As shown in Figure 8A, the RMSD reflecting the magnitude of the protein motion equilibrated at 20 ns, indicating a more stable state after binding. The decrease in the radius of gyration (*R_g_*) implies a more compacted complex structure. Overall, the simulation reaches equilibrium after 20 ns. Free energies of the amino acid residues with a distance less than 6 Å from the protein were plotted in Figure 8A, among which residues I171, M177, and V204 exhibited the lowest binding energy, indicating that they are highly involved in the binding. Besides, hydrogen bonding occupancy (Figure 8A) showed that residues E172, H173, G202, and V204 are highly involved in hydrogen bonding to isoginkgetin [36]. In the representative stereo structure of the 11S–isoginkgetin complex shown in Figure 8B, the backbone amide nitrogen of E172 formed a hydrogen bond with the carbonyl oxygen 1 in the C ring of isoginkgetin. The backbone amide nitrogen of G202 formed a hydrogen bond with the carbonyl oxygen 2 in C’ ring. This result indicated the crucial role of carbonyl groups in 11S binding. Among the top five compounds, isoginkgetin is the only one with two carbonyl groups except for pentagalloylglucose. The low binding of pentagalloylglucose to SPI could be due to the steric hindrance caused by the high molecular weight. To further investigate the major driven forces for isoginkgetin–11S interaction, we calculated 200 frames at 30–50 ns using gmx_MMPBSA. Binding free energies for the complexes are shown in Table 3. The total binding energy is −34.64 kcal/mol. Van der Waals interaction is the main driven force of isoginkgetin–11S binding, which is up to −64.15 kcal/mol. Electrostatic interactions and nonpolar solvation are also involved in the binding.

## 4. Conclusions

SPI interaction is highly dependent on the chemical structure, with a high tendency for polyphenol binding. We found that isoginkgetin has the highest apparent binding constant (*K_a_*) of (0.060 ± 0.020) × 10^6^ L·mol^−1^, followed by avicularin ((0.058 ± 0.010) × 10^6^ L·mol^−1^), pentagalloylglucose ((0.049 ± 0.010) × 10^6^ L·mol^−1^), narcissoside ((0.0013 ± 0.0004) × 10^6^ L·mol^−1^) and poliumoside ((0.0012 ± 0.0006) × 10^6^ L·mol^−1^). The interaction between soybean 11S glycinin and isoginkgetin is mostly driven by van der Waals force. Protein residues E172, H173, G202, and V204 are highly involved in hydrogen bonding with carbonyl oxygens of isoginketin, which could be the crucial events in SPI binding. Our study located isoginkgetin as a novel SPI binder, which could be a potential modifier used in SPI-based emulsions or gels, likely in the manufacture of plant-based drinks and meat analogs. Our study also provides a research strategy combining virtual screening, fluorescence quenching, and MD simulation, to characterize plant protein and phytochemicals interaction, which has important implications for the utilization of SPI–phytochemical complex in the food matrix.

## Figures and Tables

**Figure 1 foods-12-00272-f001:**
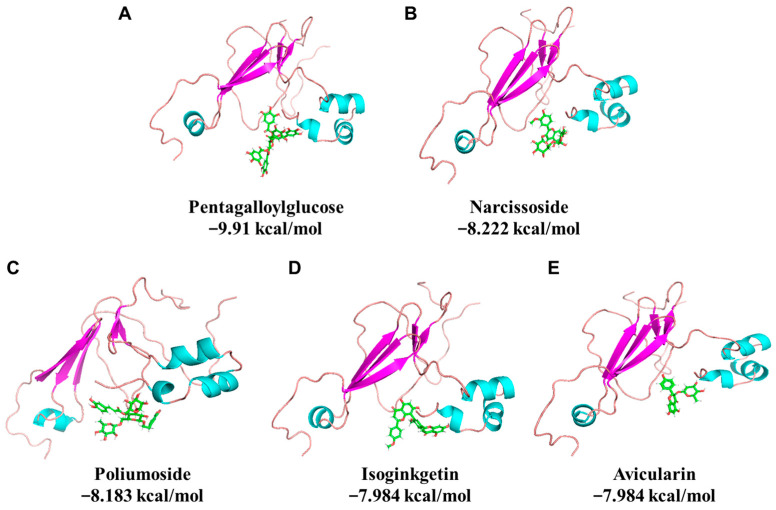
The stereo views of leading phytochemical–11S complexes from virtual screening. (**A**) Pentagalloylglucose, (**B**) narcissoside, (**C**) poliumoside, (**D**) isoginkgetin, (**E**) avicularin.

**Figure 2 foods-12-00272-f002:**
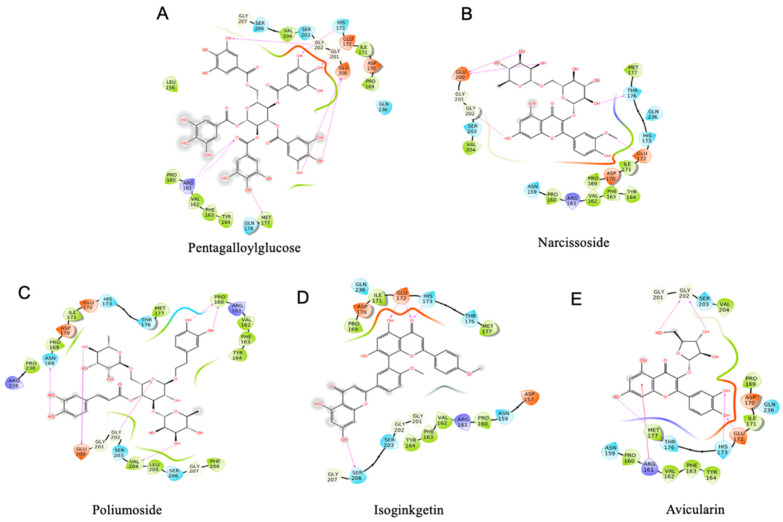
Molecular interaction diagram of top-ranking phytochemical binding with soybean 11S globulin. (**A**) Pentagalloylglucose, (**B**) narcissoside, (**C**) poliumoside, (**D**) isoginkgetin, (**E**) avicularin.

**Figure 3 foods-12-00272-f003:**
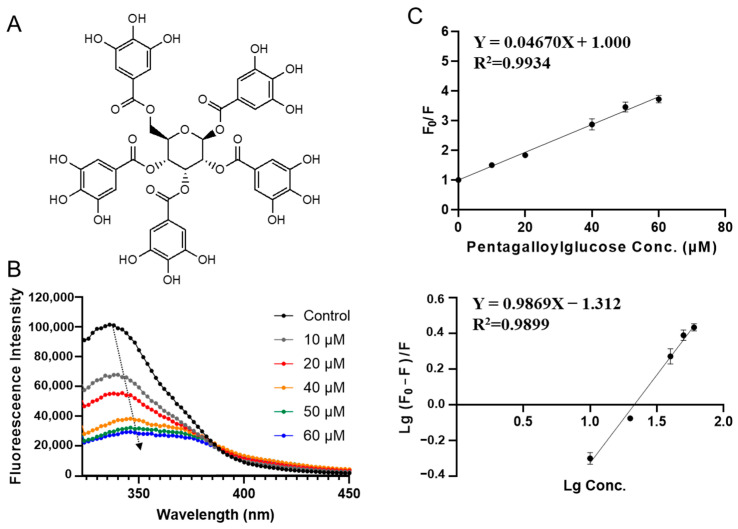
Characterization of SPI–pentagalloylglucose interaction by fluorescence quenching assay. (**A**) Molecular chemical structure of pentagalloylglucose. (**B**) The fluorescence emission spectra of SPI titrated with different concentrations of pentagalloylglucose (10, 20, 40, 50, 60 μΜ). (**C**) The Stern–Volmer plots for binding parameter calculation.

**Figure 4 foods-12-00272-f004:**
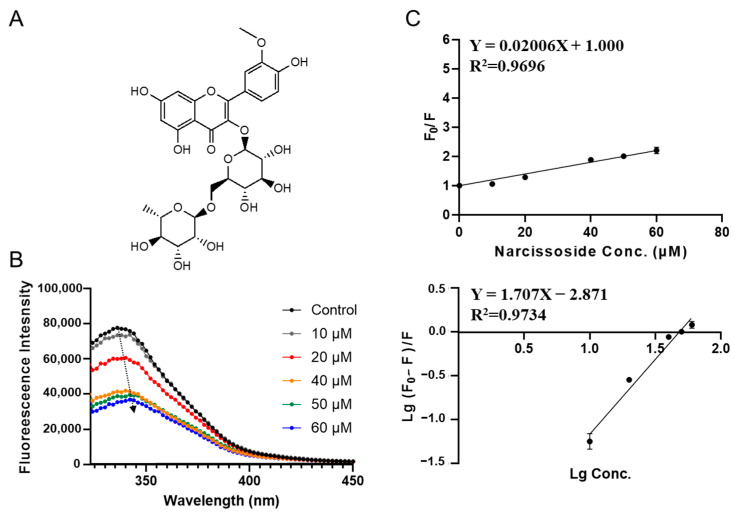
Characterization of SPI–narcissoside interaction by fluorescence quenching assay. (**A**) Molecular chemical structure of narcissoside. (**B**) The fluorescence emission spectra of SPI titrated with different concentrations of narcissoside (10, 20, 40, 50, 60 μΜ). (**C**) The Stern–Volmer plots for binding parameter calculation.

**Figure 5 foods-12-00272-f005:**
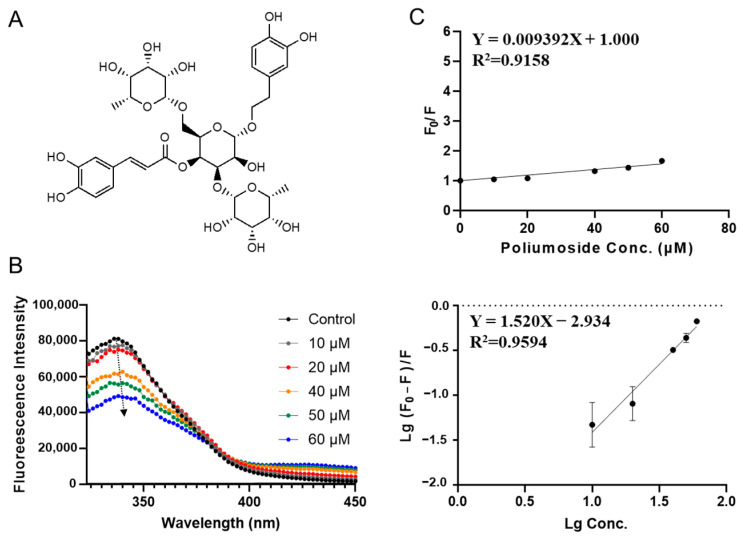
Characterization of SPI–poliumoside interaction by fluorescence quenching assay. (**A**) Molecular chemical structure of poliumoside. (**B**) The fluorescence emission spectra of SPI titrated with different concentrations of poliumoside (10, 20, 40, 50, 60, μΜ). (**C**) The Stern–Volmer plots for binding parameter calculation.

**Figure 6 foods-12-00272-f006:**
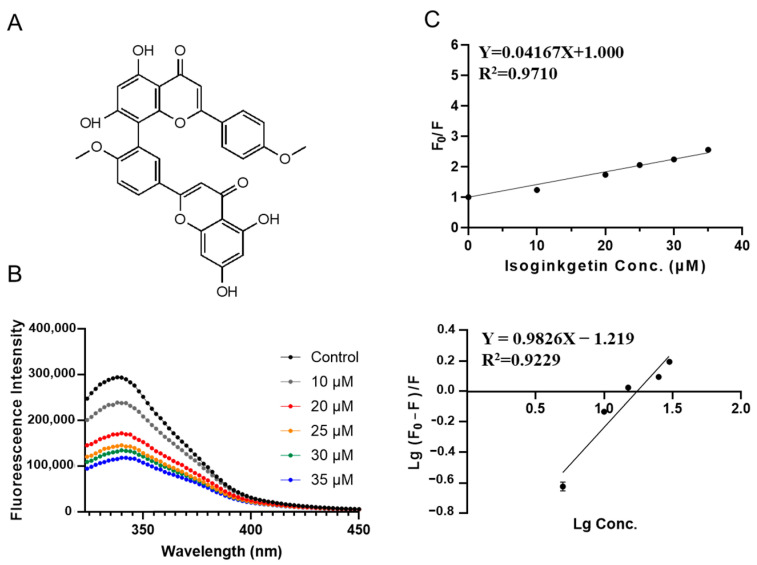
Characterization of SPI–isoginkgetin interaction by fluorescence quenching assay. (**A**) Molecular chemical structure of isoginkgetin. (**B**) The fluorescence emission spectra of SPI titrated with different concentrations of isoginkgetin (10, 20, 25, 30, 35 μΜ). (**C**) The Stern–Volmer plots for binding parameter calculation.

**Figure 7 foods-12-00272-f007:**
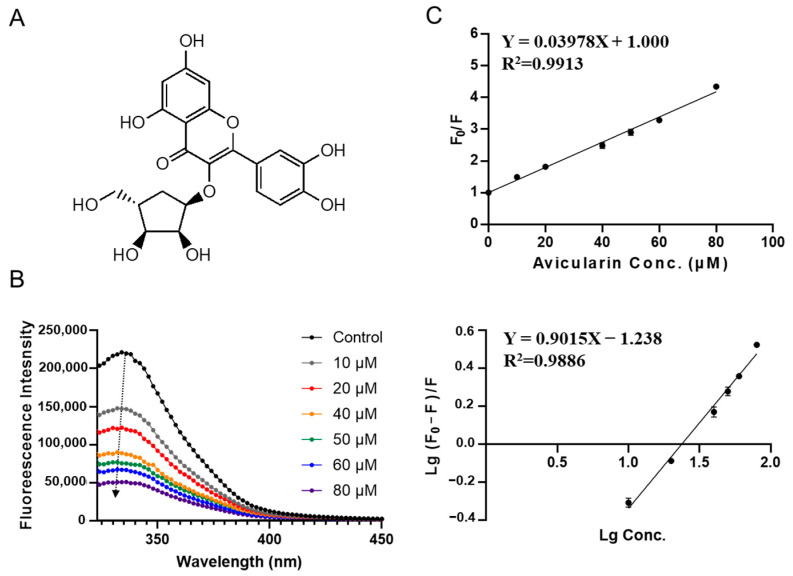
Fluorescence spectroscopy of SPI–avicularin interaction. (**A**) Molecular structure of avicularin and fluorescence spectra of SPI at different concentrations of avicularin (10, 20, 40, 50, 60, 80 μΜ). (**B**) The Stern–Volmer curve chart. (**C**) The Stern-Volmer plots for binding parameter calculation.

**Figure 8 foods-12-00272-f008:**
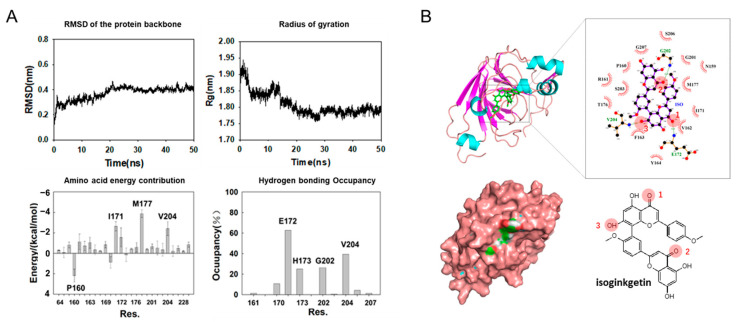
MD simulation of SPI and isoginkgetin interaction. (**A**) RMSD, radius of gyration, hydrogen bonding occupancy and amino acid energy contribution during simulation. (**B**) The representative stereo structure of 11S–isoginkgetin complex.

**Table 1 foods-12-00272-t001:** Top 20 phytochemical compounds from virtual screening.

Compounds	CAS	Molecular Weight (Da)	Docking Energy (kcal/mol)	Ranking
Pentagalloylglucose	14937-32-7	940.7	−9.91	1
Narcissoside	604-80-8	624.5	−8.222	2
Poliumoside	94079-81-9	770.7	−8.183	3
Isoginkgetin	548-19-6	566.5	−7.984	4
Avicularin	572-30-5	434.3	−7.984	5
Isoliensinine	6817-41-0	610.7	−7.938	6
Isoquercitrin	482-35-9	464.4	−7.913	7
Thonningianin A	271579-11-4	874.7	−7.823	8
2′-O-galloylhyperin	53209-27-1	616.5	−7.749	9
Forsythoside B	81525-13-5	756.7	−7.743	10
Crotonoside	1818-71-9	283.24	−7.727	11
Gnetol	86361-55-9	244.24	−7.7	12
Oxyresveratrol	29700-22-9	244.24	−7.682	13
(-)-swainsonine	136997-64-3	712.7	−7.639	14
Amentoflavone	1617-53-4	538.5	−7.584	15
Brazilin	474-07-7	286.28	−7.538	16
Liquiritin Apioside	74639-14-8	550.5	−7.491	17
Rhodiosin	86831-54-1	610.5	−7.489	18
Chelidonine	476-32-4	353.4	−7.452	19
Rhapontigenin	500-65-2	258.269	−7.442	20

**Table 2 foods-12-00272-t002:** Binding parameters of selected SPI–phytochemical compound interaction.

Compounds	*K_sv_* (×10^6^ L·mol^−1^)	*K_a_* (×10^6^ L·mol^−1^)	*n*
Pentagalloylglucose	0.047 ± 0.001 ^a^	0.049 ± 0.010 ^a^	1.00 ± 0.04 ^b^
Narcissoside	0.020 ± 0.001 ^c^	0.0013 ± 0.0004 ^b^	1.71 ± 0.16 ^a^
Poliumoside	0.0094 ± 0.0010 ^d^	0.0012 ± 0.0006 ^b^	1.52 ± 0.18 ^a^
Isoginkgetin	0.042 ± 0.002 ^b^	0.060 ± 0.020 ^a^	0.98 ± 0.12 ^b^
Avicularin	0.040 ± 0.001 ^b^	0.058 ± 0.010 ^a^	0.90 ± 0.05 ^b^

^a, b, c, d^ Significant difference in values with different letters in the same column.

**Table 3 foods-12-00272-t003:** Binding energy contribution of isoginkgetin–11S complex.

Title 1	Title 2
Van der Waals	−64.15
Electrostatic	−8.13
Polar solvation	42.46
Nonpolar solvation	−4.82
*G_gas_*	−72.28
*G_solv_*	37.64
Total	−34.64

Notes: *G_gas_* = van der Waals + electrostatic, *G_solv_* = polar solvation + nonpolar solvation, Total = *G_gas_* + *G_solv_*.

## Data Availability

The data presented in this study are available on request from the corresponding author.

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
