# Peer review of "Virtual Screening of Soybean Protein Isolate-Binding Phytochemicals and Interaction Characterization"

_foods, 2023, doi:10.3390/foods12020272_

Round 1
Reviewer 1 Report
The article is interesting, the novelty of the study is clearly stated. The following recommendations are suggested.
Abstract. OK
Introduction. OK
Materials & Methods. OK
Results. Avoid the repetition of results already included in Tables and Figures, it is redundant.
Discussion. OK
Conclusion. Authors must avoid a summary of the results in this section, but include only the concluding remarks.
Author Response
Dear Reviewer:
Thank you for your valuable comments concerning our manuscript. Those comments are all very helpful for improving our paper, as well as of important guiding significance for our future research.
We've modified the description of results, avoiding those already included in Tables and Figures. We've also shorten the Conclusion and only included the concluding remarks.
Thank you again for your comments.
Reviewer 2 Report
The article submitted for review entitled Virtual Screening of Soybean Protein Isolate-Binding Phyto-2 chemicals and Interaction Characterization, concerns a very interesting area of searching for factors responsible for the binding potential of nanoparticles.
The structure of the article, the literature review, the selection of methods and the description of the results are correct and do not raise any doubts.
The study did not notice an excessive number of self-citations, and the literature used on the subject does not raise any doubts.
The presentation of the manuscript is clear and unambiguous.
However, during the evaluation of the article submitted for review, several shortcomings were noticed:
1. No clear purpose of research. Although possible interactions are explained in the introduction, there is no reference to real conditions in the construction of the experiment. Perhaps it is worth confronting the described experiences with real experiences that would confirm/validate the observed relationships.
2. In fact, in the conclusions there is no substantive (because there was no purpose of the research) reference of the results to the conditions in the foodstuffs or the potential of their use.
In the reviewer's opinion, the article in its current form is very valuable, but I have doubts whether it is suitable for publication in a food journal, because there are no direct trials of application. The forces and results indicated in the description may apply to many other applications.
Due to the lack of a precise purpose and indicated application sense, I believe that the article should not be published in FOODS, but should go to a biochemical journal, e.g. Molecules.
Reviewer 3 Report
Manuscript: foods-2080832
Virtual Screening of Soybean Protein Isolate-Binding Phyto- 2 chemicals and Interaction Characterization
This manuscript reports high throughput virtual screening method to study on a phytochemical library (1130 compounds) to pinpoint potential soy protein isolate (SPI) binding molecules. Top-five ranking molecules from screening were found to be pentagalloylglucose, narcissoside, poliumoside, isoginkgetin and avicurin. This study indicated that SPI interaction is highly dependent on the chemical structure, with high tendency for polyphenol binding.
A number of advanced techniques were used to study and characterize phytochemicals in SPI including fluorescence quenching assays. Moreover, MD simulation technique demonstrated that the interaction between soybean 11S glycinin and isoginkgetin is mostly driven by Van der Waals force. Overall, this study suggests a research strategy combining virtual screening, fluorescence quenching and MD simulation, to characterize plant protein and phytochemicals interaction, which has important implications for the utilization of SPI-phytochemical complex in food matrix.
In my opinion this manuscript has enough novelty to be published in Foods in its current form. However, I only suggest some following minor corrections:
Abstract: The format of abstracts needs to be checked to cover the requirements with MDPI-Foods. It seems that you do not need to start with numerical and phrases like Background, ….
Introduction: You may extend your introduction by including most recent literature close to your research theme and explain your research hypothesis and novelty. For example, 2nd paragraph of your introduction needs to be extended to cover above mentioned points.
Materials and Methods: OK
Results and Discussion: All figures and tables are very well organized and discussed.
Conclusions: In this section you do not need to cite any references. Please explain the main findings of the study.
Author Response
Response to Reviewer 3
Dear Reviewer:
Thank you for your valuable comments concerning our manuscript. Those comments are all very helpful for improving our paper, as well as of important guiding significance for our future research. We have studied your comments carefully and have made correction which we hope to meet with your approval. Revised portion are all highlighted in red in the revised manuscript. The main corrections in the paper and the responses to your comments are listed as following.
1、Abstract: The format of abstracts needs to be checked to cover the requirements with MDPI-Foods. It seems that you do not need to start with numerical and phrases like Background, ….
Response:We’ve modified the abstract to avoid start with headings like Background…
2、Introduction: You may extend your introduction by including most recent literature close to your research theme and explain your research hypothesis and novelty. For example, 2nd paragraph of your introduction needs to be extended to cover above mentioned points.
Response: We’ve explained more about the research hypothesis and novelty of the research in the Introduction section. Please see Lines 34-41, Lines 46-56, and Lines 58-63 in the revised MS.
3、Conclusions: In this section you do not need to cite any references. Please explain the main findings of the study.
Response:We’ve deleted the references in the Conclusion section and added the main findings of the study.
Reviewer 4 Report
Dear Authors,
The manuscript by Panhang Liu and co-workers presents the virtual screening of soybean protein isolate-binding phytochemicals. The manuscript was quite carefully prepared, although some linguistic errors can be found. Therefore, I believe that the English language should be checked carefully. Virtual screening is currently an interesting and developing topic, so I believe this topic will interest a wide range of readers. However, I have some points that must be addressed before the paper is suitable for publication. So, this manuscript should be accepted after major revision.
In details:
(1) The English language should be carefully rechecked. There are many errors in the manuscript.
(2) In the Method part, the Authors wrote in line 73: "A virtual screening was performed on a phytochemical database with 1130 compounds, using Schrödinger Glide based on the grid files generated previously. The phytochemical compounds were docked into the optimum pocket and scored by the ligand docking function."
More detailed information regarding virtual screening should be provided.
The authors wrote that the "grid files generated previously" What does this mean?
What do the authors mean when they say the ligands were docked in the optimal pocket? What is the optimal pocket? This must be supplemented.
(3) Equations 1 and 2 imply that Q is the concentration of FREE, unbound quencher, but the authors used TOTAL concentration in their analyses. This makes no difference if the quencher is in great excess over protein. I am not sure if these conditions are met here. What was the ratio of the concentrations of the used compounds to the protein? If the excess is not large, other equations must be used. Moreover, the analyses presented in Table 2 will not be valid in these circumstances.
(4) In my opinion, the authors should also select at least one compound that turned out to be unpromising from virtual screening and also determine the binding constant (Ka) for it - this will be a negative control in fluorescence studies.
(5) Were the results obtained in the fluorescence tests corrected for the effect of the internal filter? I think not. I have not found such information. This correction must be made. All analyzes performed in fluorescence tests will then be changed.
Round 2
Reviewer 2 Report
I accept the authors' corrections and have no further comments
Reviewer 4 Report
The authors responded to all comments in a satisfactory manner. Therefore, I believe that the manuscript can be published in its present form.